# Prospective Observational Study of COVID-19 Vaccination in Patients with Thoracic Malignancies: Adverse Events, Breakthrough Infections and Survival Outcomes

**DOI:** 10.3390/biomedicines12030535

**Published:** 2024-02-27

**Authors:** Urska Janzic, Andrej Janzic, Abed Agbarya, Urska Bidovec-Stojkovic, Katja Mohorcic, Marina Caks, Peter Korosec, Matija Rijavec, Erik Skof

**Affiliations:** 1Department of Medical Oncology, University Clinic Golnik, 4204 Golnik, Slovenia; katja.mohorcic@klinika-golnik.si; 2Medical Faculty Ljubljana, University of Ljubljana, 1000 Ljubljana, Slovenia; eskof@onko-i.si; 3Faculty of Medicine, University of Maribor, 2000 Maribor, Slovenia; andrej.janzic@outlook.com; 4Bnai-Zion Medical Center, Oncology Institute, Haifa 31048, Israel; abed.agbarya@b-zion.org.il; 5Laboratory for Clinical Immunology and Molecular Genetics, University Clinic of Respiratory and Allergic Diseases Golnik, 4204 Golnik, Slovenia; urska.bidovec-stojkovic@klinika-golnik.si (U.B.-S.); peter.korosec@klinika-golnik.si (P.K.); matija.rijavec@klinika-golnik.si (M.R.); 6Department of Oncology, University Medical Centre Maribor, 2000 Maribor, Slovenia; marina.caks@ukc-mb.si; 7Faculty of Pharmacy, University of Ljubljana, 1000 Ljubljana, Slovenia; 8Biotechnical Faculty, University of Ljubljana, 1000 Ljubljana, Slovenia; 9Department of Medical Oncology, Institute of Oncology Ljubljana, 1000 Ljubljana, Slovenia

**Keywords:** thoracic malignancies, cancer therapy, COVID-19 vaccination, adverse events, breakthrough infection

## Abstract

Due to the devastating COVID-19 pandemic, a preventive tool in the form of vaccination was introduced. Thoracic cancer patients had one of the highest rates of morbidity and mortality due to COVID-19 disease, but the lack of data about the safety and effectiveness of vaccines in this population triggered studies like ours to explore these parameters in a cancer population. Out of 98 patients with thoracic malignancies vaccinated per protocol, 60–75% experienced some adverse events (AE) after their first or second vaccination, most of them were mild and did not interfere with their daily activities. Out of 17 severe AEs reported, all but one were resolved shortly after vaccination. No significant differences were noted considering AE occurrence between different cancer therapies received after the first or second vaccination dose, *p* = 0.767 and *p* = 0.441, respectively. There were 37 breakthrough infections either after the first (1), second (13) or third (23) vaccine dose. One patient died as a direct consequence of COVID-19 infection and respiratory failure, and another after disease progression with simultaneous severe infection. Eight patients had moderate disease courses, received antiviral therapies and survived without consequences. Vaccination did not affect the time to disease progression or death from underlying cancer.

## 1. Introduction

The COVID-19 pandemic hit the world like an asteroid in early 2020, leaving us in a state of unprepared emergency and crisis. By today, more than 700 million people have been infected with the SARS-CoV-2 virus, and almost 7 million have died from complications of infection [1]. At the beginning of pandemics, virus transmission and mortality were reduced by a range of precautions such as social distancing, restricted contacts, hand hygiene, facemasks and even more radical solutions such as lockdown of schools, governmental and non-essential facilities. All that had little effect on COVID-19 contraction and set the scientists in the race for effective vaccine development [2]. Indeed, the first vaccines were developed rapidly and presented an exceptional and elegant solution to the problem [3,4,5,6,7]. 

Only a year after the pandemic was declared, vaccines passed the trials and were introduced to the general population [2]. Since vaccine production could not meet high demand, some priority groups were set to receive vaccination amongst the first, and cancer patients were definitely one of them [8]. Patients with thoracic malignancies had an increased risk of COVID-19 infection, morbidity and mortality due to their underlying disease, smoking status, pulmonary damage, immunocompromised state and cancer treatment [9]. Initial reports have revealed that as much as 60–75% of SARS-CoV-2 infected patients with thoracic malignancies needed hospitalization due to severe COVID-19 disease course, were less likely to receive intensive care unit treatment if needed and that the infection resulted in a devastating 25–33% mortality [8,10]. Since data on the efficacy and safety of COVID-19 vaccines were scarce concerning cancer populations, many oncologic societies and individuals set up trials assessing not only the rate of seroconversion and protection but also safety concerning adverse events amongst cancer patients who were vaccinated [11,12,13,14]. Initial concerns were directed especially towards cancer patients treated with chemotherapy, which might have a detrimental effect on vaccine production, but also towards immune checkpoint inhibitor therapy, which could, in theory, overstimulate patients’ immune systems and expose them to a much higher risk of adverse events. Since cancer patients on active treatment were mostly excluded from vaccination trials, the oncological society was in an unenviable position, with the desire to protect cancer patients and to carry out lifesaving or at least life-prolonging treatments on one hand, and on the other hand, a potential protective agent in the form of a vaccine, for which we had no real data about how it would fit in the cancer treatment scenario. 

Therefore, we conducted a prospective observational study (PRO-ONCO-COVID-19) to assess both the immunogenicity and rate of protection, but also the safety and possible adverse effects after vaccination in solid cancer patients being treated for their disease. We have already reported the immunogenicity and antibody production of cancer patients after the primary course of vaccination and after the third booster dose [15,16]. More data on the safety and outcomes of the primary disease for which patients have been treated are warranted for better knowledge of how to act in these situations in the future. Here we present data on safety, adverse events after vaccination, breakthrough COVID-19 infections and outcomes of both infection and underlying disease for patients with thoracic malignancies, which comprised the largest cohort in our study.

## 2. Materials and Methods

### 2.1. Study Design and Participants

We conducted a prospective observational multicentric study of cancer patients on systemic cancer treatment (or within a year of the last dose of therapy received) who were voluntarily vaccinated against COVID-19. All patients signed written consent before study inclusion. Demographic data and data on treatment were collected at the time of informed consent signing. The study and its amendment after the third dose of vaccine were approved by the Slovenian National Ethics Committee (No. 0120-39/2021/6 and 0120-39/2021/9). This trial was conducted in accordance with the ethical principles of the Declaration of Helsinki.

### 2.2. Procedures 

Patients were treated with systemic cancer therapy for thoracic malignancies (non-small cell lung cancer, small cell lung cancer and malignant mesothelioma) either at the moment of first vaccination or within a year prior to receiving the first vaccine dose in two Slovenian academic institutions. They were also vaccinated with either mRNA vaccines BNT162b2 (Pfizer, New York, NY, USA/BioNTech, Germany, EU) or mRNA-1273 (Moderna, Cambridge, MA, USA) or adenoviral AZD1222 (AstraZeneca, Cambridge, UK) vaccines per manufacturing instructions for the primary course of vaccination and all of the patients received one of the mRNA vaccines for the third booster dose of vaccination as per National Vaccination Strategy recommendations. Data about blood sampling and anti-SARS-CoV-2 S1 IgG antibody detection were described before in detail in previous publications [15,16]. 

### 2.3. Recognition of Adverse Events

Adverse events (AEs) were evaluated during telephone consultations carried out by highly trained coordinators 24–48 h after each vaccination dose of the primary course of vaccination to assess possible safety concerns and AEs and were recorded on pre-prepared forms. AEs were graded with grades (G) ranging from 0 to 4 (G0 = no AEs; G1 = mild AEs; G2 = moderate AEs that do not interfere with daily activities; G3 = severe AEs that interfere with daily activities; G4 = life-threatening AEs, hospitalization required). The AEs evaluated were local (pain, redness or swelling on injection site) or systemic (fever, chills, fatigue, myalgia, arthralgia, headache, vomiting, diarrhea, other) in line with trials of mRNA vaccines [3,4]. All AEs were reported by the national pharmacovigilance system to the National Institute for Public Health, as appropriate.

### 2.4. Detection of Breakthrough COVID-19 Infection 

Prior COVID-19 infection was not an exclusion factor but was documented and reported, as well as all of the cases of breakthrough COVID-19 infections, with date and severity, according to WHO and NIH guidelines [9]. Breakthrough infections were considered as: mild (symptoms of acute upper respiratory tract infection without pneumonia), moderate (radiological evidence of pneumonia), severe (pneumonia with dyspnea and hypoxemia) and critical (pneumonia with acute respiratory distress syndrome, respiratory failure, shock and multiple organ dysfunction) [9]. In case of severe distress (clinical signs of pneumonia—fever, cough, dyspnoea, plus one of the following: respiratory rate > 30 breaths/min; severe respiratory distress; or SpO2 < 90% on room air) patients were hospitalized and treated accordingly. 

### 2.5. Statistical Analysis and Outcomes

A safety analysis of the occurrence of AEs following the primary course of vaccination was compared between different therapies received with the Chi-squared test. The proportion of patients with SARS-CoV-2 infection after the complete primary course of vaccination or after the third booster dose was described and presented according to G (grades) of disease severity. The median overall survival of patients with thoracic malignancies was estimated according to disease and therapy specifics using Kaplan–Meier method and was calculated from the time of diagnosis until death. The R Studio build 494 software (Boston, MA, USA), was used for all statistical analyses. 

## 3. Results

There were 125 patients with solid cancers included in the overall study, 98 of whom were treated for thoracic malignancies (87 non-small cell lung cancer (NSCLC), 6 small cell lung cancer (SCLC), 5 malignant mesotheliomas) with a slight male preponderance of 55% and a median age of 63.5 years at the time of diagnosis. Most of the patients (91%) were receiving therapy for metastatic disease and a similar proportion (90%) were on active treatment at the time of the first vaccination. The patients were receiving either chemotherapy, chemotherapy plus immune checkpoint inhibitors (ICI), ICI alone or targeted therapy in 16%, 20%, 28% and 36%, respectively. Most of the patients were vaccinated with the mRNA-based BNT162b2 (Pfizer/BioNTech) vaccine for the primary course of vaccination and all of the patients that opted for booster vaccination received that vaccine as the third dose. More details on patients and their characteristics are in Table 1.

### 3.1. Adverse Events after the Primary Course of Vaccination

After the first and second vaccine doses, 60% and 75% experienced local and 35% and 22% systemic adverse events, respectively. Altogether, 36% of patients experienced adverse events after both vaccination doses. The most common local adverse event was moderate pain on the injection site in 60% and 70%, and the most common systemic adverse events were fatigue in 23% and 28%, headache in 10% and 13% and musculoskeletal pain in 11% and 17% after first and second vaccine dose, respectively. 

After the first vaccination dose, the majority of patients (97%) experiencing local AEs described them as mild (G1). Patients experiencing both local and systemic AEs reported them as moderate (G2) in 8% and severe in 3%. One patient had a severe (G4) life-threatening AE that required hospitalization. No differences in the occurrence of AEs were observed between treatment groups (*p* = 0.767). After the second vaccine dose, twenty patients (21%) and three patients (3%) reported G2 and G3 AEs, respectively. Again, there were no statistically significant differences between treatment groups in terms of local (*p* = 0.978) or systemic (*p* = 0.441) AEs. A detailed representation of adverse events regarding cancer therapy received is presented in Figure 1. Patients treated with ICI had a numerically higher frequency of musculoskeletal adverse events, such as myalgia and arthralgia, which was not statistically significant and can also be attributed to the type of cancer therapy received. 

Seventeen patients experienced severe adverse events that interfered with their daily activities or even required medical assistance (Figure 2). Interestingly, the same two patients with serious (G3) AEs after the first vaccine dose also reported G3 AEs after the second vaccine dose, but the type of AE was different. The first patient reported serious myalgia and fatigue and the second severe local pain and headache after the first and second vaccine doses, respectively. One of the patients was diagnosed with G4 pericarditis and myocarditis and was hospitalized for a prolonged period of time and required intensive therapy after the first vaccine dose, which was the only vaccine dose that he received.

Of note, that patient was treated with ICI at the time of AE emergence and the causality of the adverse event was not finally determined. Also, the patient survived another 8 months after the vaccination and the severe AE emergence.

### 3.2. Breakthrough COVID-19 Infections

There were 37 breakthrough COVID-19 infections detected amongst our cohort of patients with thoracic malignancies vaccinated against SARS-CoV-2. Of those, 1, 13 and 23 patients received one, two and three vaccine doses, respectively. The patient who received a single vaccine dose was treated for metastatic non-small cell lung cancer with ICI therapy, contracted COVID-19 eight days after the vaccination and had a critical disease course. He was treated in the intensive care unit with oxygen supplementation, antibiotics and dexamethasone, but died 14 days after the infection as a result of pneumonia with acute respiratory distress syndrome. The predominant variant of concern in Slovenia at that time was SARS-CoV-2 alpha [17]. 

Out of the 13 patients that received two doses of vaccine and completed the primary course of vaccination, the breakthrough infection happened in a median of 230 days (range 35–510 days) after the second vaccine dose. Seven patients had mild disease course, four patients moderate—of those, three received antiviral therapy with casirivimab + imdevimab or sotrovimab as an outpatient treatment and one patient had severe disease course and was hospitalized, needed oxygen supplementation, received remdesivir and survived without severe consequences. The last patient was treated for metastatic small cell lung cancer and contracted the virus 220 days after the second vaccine dose. He had a critical disease course, was hospitalized and received supportive measures and both remdesivir and dexamethasone, but died because of the combination of cancer progression and respiratory failure due to severe COVID-19 infection. 

The 23 patients who received three vaccination doses were infected with SARS-CoV-2 in a median of 230 days (range 6–469 days) after the last vaccine dose. Most of the patients (18/22) had only a mild disease course, four had a moderate disease course and were treated with antiviral drugs in an outpatient setting (one casirivimab + imdevimab, one sotrovimab and two remdesivir) and one patient had severe COVID-19 infection and was treated in a hospital with supportive measures, oxygen and methylprednisolone, and recovered completely. Originally, he was treated with targeted therapy for metastatic non-small cell lung cancer and is still alive at the data cut-off. Of note, SARS-CoV-2 delta and omicron were the VOCs in Slovenia at that time [17].

### 3.3. Thoracic Malignancies Treatment Outcomes

The median time for the whole cohort of patients with thoracic malignancies from any cancer treatment starting from the date of first vaccination was 13.8 months (range 0.5–71.3 months). As shown in Table 1, our cohort of patients with thoracic malignancies is comprised mainly of NSCLC patients (89%), and only a minority of SCLC and malignant mesothelioma patients (6% and 5%, respectively). At data cutoff on 1 February 2024, 50/98 patients were still alive and in follow-up or receiving treatment. 

All of the SCLC patients were treated for metastatic disease, all but one with the combination of chemotherapy and ICI and one patient with maintenance ICI alone at the time of first vaccination. At data cut-off, 4/6 patients have died, resulting in a median overall survival of 24.7 months (95% CI 15.3–NA), calculated from the start of any oncological treatment. Concerning malignant mesothelioma patients, one patient had local disease that became advanced in the course of the treatment, two had locally advanced disease and two were metastatic at the time of diagnosis. At the time of first vaccination, two patients were receiving chemotherapy and three were treated with ICI. All of the patients died by data cutoff, with a median overall survival from the onset of the first oncology therapy of 35.0 months (95% CI 23.2–NA) across all disease stages. 

Seven NSCLC patients were treated for limited-stage disease; of those, six were treated with adjuvant chemotherapy at the time of vaccination and one was already receiving a combination of chemotherapy and ICI due to disease progression. All but one patient from that group are still alive; the median observational time is 42.1 months (range 33.0–65.7 months). Out of eight patients receiving chemotherapy for metastatic disease at the time of first vaccination, half of them received it as first-line therapy due to the unavailability of chemotherapy + ICI in the first-line setting, and the other half received it as therapy for later lines of disease, where other treatment options were exhausted. Fourteen patients were receiving the combination of chemotherapy and ICI, all but two as first-line treatment for metastatic disease, and nine patients died due to disease progression at the time of data cutoff, with a median overall survival of 32.2 months (95% CI 26.9–NA). Another 23 NSCLC patients were receiving ICI alone, 70% as first-line treatment, and 14 deaths were detected at data cutoff. Median overall survival for this group of patients was 44.4 months (95% CI 31.6–NA). The last subgroup of 35 oncogene-addicted NSCLC were all receiving targeted therapy at the time of vaccination and were mostly still alive at the data cutoff (25/35 patients) with a median follow-up time of 64.5 months (range 35.6–94.5 months). The timing of vaccination did not negatively affect the time of cancer therapy receipt, neither did it cause further disease progression. Timelines of patients with thoracic malignancies receiving different treatment strategies and the timing of vaccination doses received are presented in Figure 3.

## 4. Discussion

The rate at which COVID-19 vaccines were developed and introduced to the public, was exceptional [3,4,5]. Since then, more than 67% of the world’s population has received the primary course of vaccination protecting them against disease, and altogether over 13 billion vaccines have been administered as of June 2023. Nevertheless, almost 7 million deaths were reported since the start of the pandemic, accounting for about a 1% case fatality rate [18]. 

Still, a meta-analysis of 52 studies early on in the pandemic revealed a devastating 25% case fatality rate of cancer patients in case of COVID-19 infection [19]. The most endangered population was patients with hematological malignancies, but it seems that close behind, patients with thoracic malignancies had one of the highest rates of mortality [9]. Patients with thoracic malignancies suffered from a longer and more severe course of COVID-19 because of their underlying illness, COPD and smoking damage to the lungs, resulting in 62–76% of hospitalizations and 25–33% of deaths directly attributed to COVID-19 [8,10,20,21]. Moreover, due to the unknown, cancer patients were reluctant to get vaccinated at first, fearing adverse events of vaccination, but also interaction with their oncological treatment and possible progression of disease [22,23]. 

As reported before, COVID-19 vaccination was very effective in the solid cancer population, reaching high immunogenicity with antibody production, which acted as a safety tool both protecting from infection as well as severe disease course in case of infection [13,15,24,25,26]. Adverse events were considered manageable and mostly mild. However, a lot of concern was raised at the beginning of the pandemic about treatment with immune checkpoint inhibitors and simultaneous vaccination with COVID-19 vaccines since that could elicit the patient’s immune response and cause worse ICI-related adverse events [27]. Initial large trials, which included large numbers of cancer patients, did not show concerning rates of adverse events in this patient population, even though they either included both patients with hematological and solid cancers or any type of cancer therapy received [12,13,24,28,29]. In a large prospective multicenter study where investigators also divided patients according to the cancer therapy they were receiving, there was no difference in terms of grades or frequency of local or systemic adverse events after the first or second vaccination. The most common local adverse event was local pain at the injection site which ranged from 10 to 26% after the first and 5 to 9% after the second vaccine dose. The most common systemic adverse event was fatigue, again ranging from 23 to 36% after the first and 48 to 59% after the second vaccination [24]. Our results show a similar rate of local pain at the injection site, ranging from 60 to 70%, and fatigue in 21 to 32% after both vaccination courses. Likewise, the CANVAX trial showed local adverse events in 65% (mostly pain at the injection site) and systemic side effects in 50%, but no severe adverse or allergic reaction; side effects were reported as mild or moderate in all cases [12]. A study from the UK assessed both healthy controls and cancer patients and reported much fewer local and systemic adverse events after the primary course of vaccination amongst patients with cancer [28]. Overall, cancer patients did not exhibit frequent severe adverse events after vaccination, which can probably be attributed to the tolerance that this population exhibits since they are already dealing with unpleasant symptoms from disease itself or the accompanying treatment and might also attribute them to one of those reasons. Interestingly, one report describes severe skin changes in a patient receiving immune checkpoint inhibitors (ICI) after vaccination that progressed to Steven Johnsons syndrome, yet another reports severe hepatitis G4 in a similar setting [24,28]. There was one patient in our cohort as well who developed severe pericarditis/myocarditis following the first vaccine dose. Of note, our patient was receiving immune checkpoint inhibitors at the time of vaccination, which could have also caused pericarditis/myocarditis adverse events [30]. The patient was receiving ICI therapy for 6 months before the cardiac problems ensued and she received the first dose of mRNA vaccine against SARS-CoV-2 3 weeks before the complications. She was hospitalized due to severe respiratory insufficiency and spent 14 days in the intensive care unit due to cardiac tamponade that needed to be evacuated by pericardiocentesis, and her left ventricular ejection fraction decreased severely to only 20%, thus noradrenalin, dopamine and levosimendan were administered, after which she slowly recovered. Her troponin levels remained negative, but a thromb in the left ventricle was observed by CT angiography for which she received low-molecular-weight heparin. She was treated with methylprednisolone (0.5 mg/kg) and recovered completely. Since her troponin levels remained negative and the timeline of treatment with ICI precedes the vaccination with the anti-SARS-CoV-2 vaccine, the exact cause of adverse event emergence is not completely resolved but it was concluded by the infectious disease specialist that it points more in the direction of a vaccination-related adverse event. A recent meta-analysis of cardiovascular adverse events after receiving mRNA vaccines revealed the occurrence of myocarditis in 2.7% and pericarditis in 0.4% of vaccinated patients, so either of the causes is plausible [31]. 

It is clear now that the vaccination against COVID-19 protects mainly from severe disease course and less from infection itself, since the SARS-CoV-2 virus was constantly changing and acquiring new mechanisms of resistance [32]. Our data show that about 37% of patients actually contracted the virus and 22% out of those had moderate disease course and were mainly protected with additional antiviral therapies. One patient died as a direct consequence of severe infection, and another from a combination of disease progression and severe infection. That adds up to a 2% mortality in our cohort of patients with thoracic malignancies, which is still 10–15 times lower than the initial horrific reports on their mortality rate [8]. The UK group showed on a large scale that not only the primary course of vaccination but also the third booster dose is important in protecting patients against breakthrough infections. Of note, solid cancer patients derive much more benefit in terms of active or severe infection than patients with hematological malignancies [33]. Another large Canadian study revealed that cancer patients were more likely to acquire COVID-19 infection if they were treated with either chemotherapy or anti-CD20 drugs. Also, compared to solid cancer patients, those with hematological malignancies had a two-fold increased risk of severe outcomes in the case of infection [34]. Moreover, a large CCC19 initiative that included almost 2500 patients confirmed that patients with cancer who develop breakthrough COVID-19 infections following two or three doses of mRNA vaccines had better clinical outcomes compared to an unvaccinated weighted population, with lower rates of death, ICU admission and hospitalization [35].

Surely, even after the complete primary course of vaccination and the booster dose are received, there is a waning of immunity with a fall in anti-SARS-CoV-2 IgG levels, as has been shown in multiple trials [15,33,36,37,38]. Patients that have lower levels of anti-SARS-CoV-2 IgG antibodies are more likely to acquire COVID-19 infection. Our data show breakthrough infection occurring in a median of 230 days, which is certainly a long time after the last vaccine dose is received and might contribute to infection. 

The featured thoracic malignancies patient population had a similar if not even better overall survival compared to patients included in the pivotal registrational trials of either ICI or targeted therapy. Just for example, the median overall survival of non-small cell lung cancer patients treated with first-line monotherapy with immune checkpoint inhibitors is around 26.3 months and our cohort of patients treated similarly exceeded that number by 18 months [39]. Since the groups are not uniform, we cannot make assumptions or direct comparisons, but we can safely say that the vaccination against COVID-19 did not negatively impact either cancer progression or the overall survival of this patient population. 

Our study, of course, has some limitations, such as the small number of patients assessed, lack of prospective assessment of treatment groups and equal distribution of them, lack of assessment of the neutralizing capacity of current variants of concern and absence of detailed adverse event data after the third booster vaccination. However, the advantage of the study is its prospective design, preplanned time points for communication with patients and granularity of clinical data due to local insight into the patient data. 

## 5. Conclusions

Vaccination against COVID-19 has been shown to be both safe and effective in patients with thoracic malignancies. The rate of severe adverse events is low, as is the occurrence of critical disease courses in case of infection. Thinking back at what could have been done better, one could only suggest prioritizing groups with the highest mortality rates, such as thoracic malignancy patients, for the earliest vaccination and continuation of oncological treatment. There was no tool to prepare us for the disaster that struck in early 2020 and there is probably little we can do to prevent similar desolation from happening again. Lessons learned through this time should be valued, and science should be shared in the same manner as it was throughout the pandemic to accelerate further discoveries. 

## Figures and Tables

**Figure 1 biomedicines-12-00535-f001:**
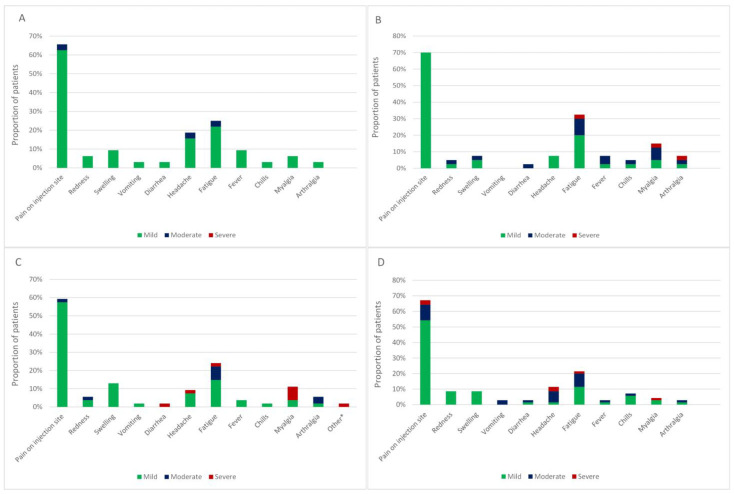
Proportion of patients with thoracic malignancies experiencing adverse events after vaccination with either first or second vaccine dose and being treated with (**A**) chemotherapy, (**B**) chemotherapy plus immune checkpoint inhibitors, (**C**) immune checkpoint inhibitor alone or (**D**) targeted therapy. * other—a case of severe pericarditis/myocarditis.

**Figure 2 biomedicines-12-00535-f002:**
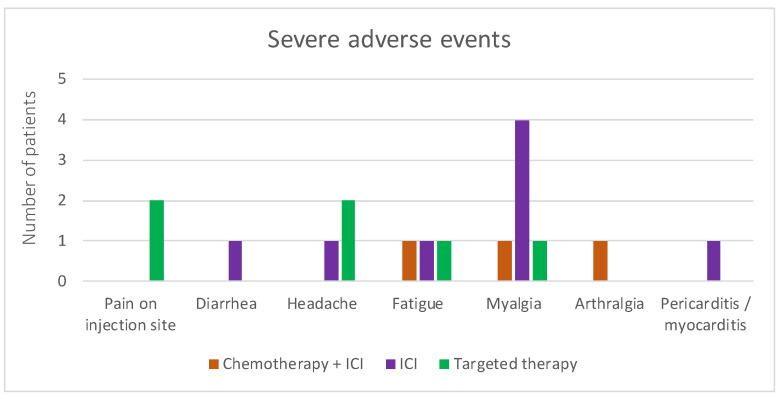
Number of patients with thoracic malignancies and active cancer treatment receiving COVID-19 vaccination and reporting severe (G3) adverse events after vaccination and one G4 AE—pericarditis/myocarditis. Orange—patients receiving chemotherapy + immune checkpoint inhibitor; purple—patients receiving immune checkpoint inhibitor alone; green—patients receiving targeted therapy.

**Figure 3 biomedicines-12-00535-f003:**
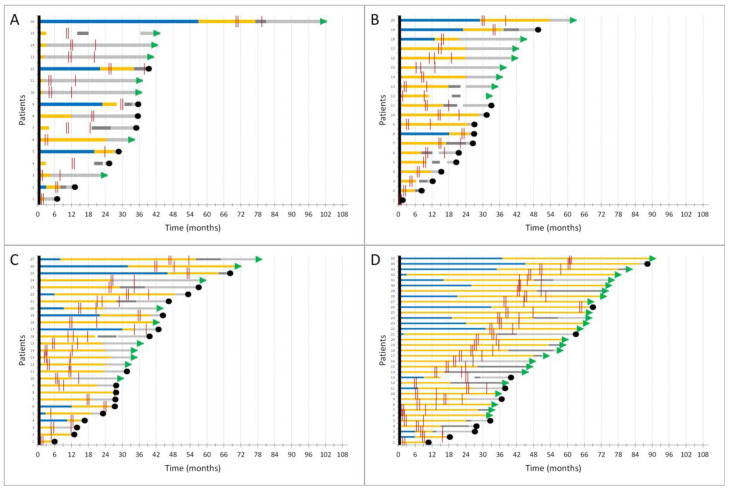
Swimmer plots of patients with thoracic malignancies receiving cancer therapy (**A**) chemotherapy, (**B**) chemotherapy + immune checkpoint inhibitors, (**C**) immune checkpoint inhibitors alone, (**D**) targeted therapy; blue lines—timeline of therapy received prior to the vaccination; yellow lines—timeline of cancer therapy received at the time of first vaccination; grey lines—further cancer therapies after disease progression; dotted line—therapy—free interval; red stripes—vaccine doses received by patients in regards to cancer therapy; green arrows represent patients still alive at data cutoff and black dots represent patients that died.

**Table 1 biomedicines-12-00535-t001:** Demographic data of patients with thoracic malignancies vaccinated for SARS-CoV-2.

	N = 98
Age in years, median (range)	63.5 (24–81)
Sex, *n* (%)	
Male	54 (55%)
Female	44 (45%)
Cancer type, *n* (%)	
NSCLC *	87 (89%)
SCLC **	6 (6%)
MPM ***	5 (5%)
Stage, *n* (%)	
Limited	8 (8%)
Locoregionally advanced	2 (2%)
Metastatic	88 (90%)
Anticancer therapy, *n* (%)	
Chemotherapy	16 (16%)
Chemotherapy + ICI	20 (20%)
ICI alone	27 (28%)
Targeted therapy	35 (36%)
Receiving systemic therapy at the time of vaccination, *n* (%)	
Yes	88 (90%)
No	10 (10%)
Positive SARS-CoV-2 IgG antibodies prior to vaccination, *n* (%) †	
No	79 (81%)
Yes	19 (19%)
No. of vaccine doses received	
First dose	98 (100%)
Second dose	95 (97%)
Third dose	57 (58%)
Type of primary vaccination received, *n* (%)	
mRNA-based BNT162b2 (Pfizer/BioNTech)	91 (91%)
mRNA-based mRNA-1273 (Moderna)	3 (4%)
Vector-based vaccine AZD1222 (AstraZeneca)	4 (5%)
Type of booster 3rd dose vaccination received, *n* (%)	
mRNA-based BNT162b2 (Pfizer/BioNTech)	57 (100%)

* NSCLC—non-small cell lung cancer; ** SCLC—small cell lung cancer; *** MPM—malignant pleural mesothelioma; †—patients were considered to have prior COVID-19 infection if they had positive SARS-CoV-2 S1 IgG antibodies above 175 ng/mL prior to first vaccination dose.

## Data Availability

https://doi.org/10.5281/zenodo.10641324 (accessed on 9 February 2024).

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
