# Peer review of "Prospective Observational Study of COVID-19 Vaccination in Patients with Thoracic Malignancies: Adverse Events, Breakthrough Infections and Survival Outcomes"

_biomedicines, 2024, doi:10.3390/biomedicines12030535_

Round 1

Reviewer 1 Report

Comments and Suggestions for Authors

This study examined adverse events and breakthrough infections after COVID-19 vaccination in 98 patients with thoracic malignancies. Key findings were that 60-75% of patients experienced mostly mild adverse events, 37 patients had breakthrough infections, and vaccination did not negatively impact cancer outcomes.

Specific comments:

1. The COVID-19 pandemic has passed its peak. Update the introduction to clarify the remaining relevance and rationale for publishing this data given the changing landscape.

2. Expand the discussion on the one case of severe pericarditis/myocarditis after vaccination and how causality was assessed.

3. Clarify the criteria for hospitalization and disease severity grading to contextualize breakthrough infection rates.

4. Discuss any waning immunity issues over time that may have impacted breakthrough rates.

5. Explain how time to cancer progression was defined and analyzed to conclude no negative impact from vaccination.

Author Response

For research article

Response to Reviewer 1 Comments

Thank you very much for taking the time to review this manuscript. Please find the detailed responses below and the corresponding revisions/corrections highlighted in the re-submitted manuscript.

Point-by-point response to Comments and Suggestions for Authors

  1. The COVID-19 pandemic has passed its peak. Update the introduction to clarify the remaining relevance and rationale for publishing this data given the changing landscape.

Response: Thank you for pointing this out. We’ve added additional rationale why we think data in this paper are still relevant. [page 2, line 74 – 76 in yellow]

  1. Expand the discussion on the one case of severe pericarditis/myocarditis after vaccination and how causality was assessed.

Response: We thank the reviewer for expressing this issue. We broadened the discussion part on this topic [page 9, line 316 – 327 in yellow]

  1. Clarify the criteria for hospitalization and disease severity grading to contextualize breakthrough infection rates.

Response: We appreciate this comment, disease severity grading is described in the Methods section under: 2.4. Detection of breakthrough COVID-19 infection: Breakthrough infections were considered as: mild (symptoms of acute upper respiratory tract infection without pneumonia), moderate (radiological evidence of pneumonia), severe (pneumonia with dyspnea and hypoxemia), and critical (pneumonia with acute respiratory distress syndrome, respiratory failure, shock, and multiple organ dysfunction). For more clarity we have added criteria for hospitalization in our center, which was: clinical signs of pneumonia - fever, cough, dyspnoea, plus one of the following: respiratory rate > 30 breaths/min; severe respiratory distress; or SpO2 < 90% on room air – these criteria were added in the Methods section. [page 3, line 119 – 122  in yellow]

  1. Discuss any waning immunity issues over time that may have impacted breakthrough rates.

Response: Thank you for pointing this out, it is certainly a relevant position and we added it in the discussion section [page 10, line 325 - 357 in yellow]

  1. Explain how time to cancer progression was defined and analyzed to conclude no negative impact from vaccination.

Response: We appreciate this comment and would like to point out that we did not calculate time to progression separately because of the different treatment modalities those patients were receiving which did not make a uniform group. Also, timelines of radiological assessment were not even across the whole group of patients, thus stating progression free survival could be deceiving. We did change the statement in the discussion section to: “Since the groups are not uniform we cannot make assumptions or direct comparisons, but we can safely say that the vaccination against COVID-19 did not negatively impact neither cancer progression, nor overall survival of this patient population. « Absence of disease progression directly after either of the vaccination course can be seen in Figure 3. [Page 9, line 360 – 365 in yellow]

Reviewer 2 Report

Comments and Suggestions for Authors

The article discusses the impact of COVID-19 on cancer patients, particularly those with thoracic malignancies, and the effectiveness and safety of COVID-19 vaccination in this population. It highlights the high mortality rate of cancer patients infected with COVID-19 and the initial concerns about vaccinating them, especially those undergoing cancer treatments. The study described in the text shows that COVID-19 vaccination was effective in generating antibody response and protecting against severe disease in cancer patients, with manageable adverse events. Despite some cases of breakthrough infections, vaccinated cancer patients had better clinical outcomes compared to unvaccinated individuals. The study also indicates that COVID-19 vaccination did not negatively impact the overall survival of patients with thoracic malignancies. However, the text acknowledges limitations in the study, such as the small sample size and the need for further assessment of vaccine effectiveness against new variants. Overall, the findings emphasize the importance of prioritizing vaccination for cancer patients and continuing oncological treatment.

The authors should respond to several question before publication:

What were the main adverse events reported after COVID-19 vaccination in cancer patients?

How did COVID-19 vaccination impact the overall survival of patients with thoracic malignancies?

What were some limitations of the study mentioned in the text?

What suggestions were made for future strategies based on the study's findings?

Comments on the Quality of English Language

English is OK

Author Response

For research article

Response to Reviewer 2 Comments

Thank you very much for taking the time to review this manuscript. Please find the detailed responses below and the corresponding revisions/corrections highlighted in the re-submitted manuscript.

Point-by-point response to Comments and Suggestions for Authors

  1. What were the main adverse events reported after COVID-19 vaccination in cancer patients?

Response: Thank you for this comment. Most of the adverse events were local, either pain, redness, or swelling at the injection site, which were experienced in 60 % after the first and 75% after the second vaccination course. Pain on the injection site was the most common local adverse event and it was experienced in 60% after first and 70% after the second vaccination course. Systemic adverse events were half as rare with 35% after the first and 22% after the second vaccination. More is visible from Figure 1.

  1. How did COVID-19 vaccination impact the overall survival of patients with thoracic malignancies?

Response: We thank the reviewer for raising this question. The vaccination did not negatively impact the overall survival of patients with thoracic malignancies. Just for example, the median overall survival of non-small cell lung cancer patients, treated with first line monotherapy with immune checkpoint inhibitors is around 26.3 months and our cohort of patients treated similarly exceeded that number by 18 months. Since the groups are not uniform we cannot make assumptions about superiority or inferiority, but we can safely say that the vaccination did not negatively impact the overall survival of patients with thoracic malignancies. [Page 9, line 360 – 365 in yellow]

  1. What were some limitations of the study mentioned in the text?

Response: We appreciate this comment. As we stated in the discussion section, our study of course, has some limitations, such as small number of patients assessed, lack of prospective assessment of treatment groups and equal distribution of them, lack of assessment of the neutralizing capacity of current variants of concern and absence of detailed adverse event data after the third booster vaccination. However, the advantage of the study is its prospective design, preplanned time points for communication with patients and granularity of clinical data due to local insight into the patient data. [page 9, line 367 - 368]

  1. What suggestions were made for future strategies based on the study's findings?

Response: We thank the reviewer fort his comment. We suggest more inclusion of cancer patients in trials that really matter, as was the case of COVID-19 vaccines. In addition, to absorb the knowledge that was gained through this pandemic and prioritize groups with the highest mortality rates, such as thoracic malignancies patients, for the earliest vaccination and continuation of oncological treatment. Vaccines against infectious diseases have shown to be effective and safe in a number of cases in our history and there is no need to fear getting the protection needed for our cancer patients, even if they are on active treatment.